# Neuro-Axonal Damage and Alteration of Blood–Brain Barrier Integrity in COVID-19 Patients

**DOI:** 10.3390/cells11162480

**Published:** 2022-08-10

**Authors:** Maria Antonella Zingaropoli, Marco Iannetta, Lorenzo Piermatteo, Patrizia Pasculli, Tiziana Latronico, Laura Mazzuti, Laura Campogiani, Leonardo Duca, Giampiero Ferraguti, Manuela De Michele, Gioacchino Galardo, Francesco Pugliese, Guido Antonelli, Massimo Andreoni, Loredana Sarmati, Miriam Lichtner, Ombretta Turriziani, Francesca Ceccherini-Silberstein, Grazia Maria Liuzzi, Claudio Maria Mastroianni, Maria Rosa Ciardi

**Affiliations:** 1Department of Public Health and Infectious Diseases, Sapienza University of Rome, 00185 Rome, Italy; 2Department of Systems Medicine, Tor Vergata University of Rome, 00133 Rome, Italy; 3Department of Experimental Medicine, Tor Vergata University of Rome, 00133 Rome, Italy; 4Department of Biosciences, Biotechnologies and Biopharmaceutics, University of Bari Aldo Moro, 70126 Bari, Italy; 5Department of Molecular Medicine, Sapienza University of Rome, 00161 Rome, Italy; 6Department of Clinical and Molecular Medicine, Sapienza University of Rome, 00161 Rome, Italy; 7Department of Experimental Medicine, Sapienza University of Rome, 00161 Rome, Italy; 8Emergency Department Stroke Unit, Department of Human Neurosciences, Sapienza University of Rome, 00161 Roma, Italy; 9Medical Emergency Unit, Sapienza University of Rome, Policlinico Umberto I, 00161 Rome, Italy; 10Department of Specialist Surgery and Organ Transplantation “Paride Stefanini”, Policlinico Umberto I, Sapienza University of Rome, 00161 Rome, Italy; 11Infectious Diseases Unit, SM Goretti Hospital, Sapienza University of Rome, 04100 Latina, Italy

**Keywords:** neurofilament light chain, NfL, matrix metalloproteinases, MMPs, ddPCR, long-COVID, neuro-COVID, zymography

## Abstract

Neurofilament light chain (NfL) is a specific biomarker of neuro-axonal damage. Matrix metalloproteinases (MMPs) are zinc-dependent enzymes involved in blood–brain barrier (BBB) integrity. We explored neuro-axonal damage, alteration of BBB integrity and SARS-CoV-2 RNA presence in COVID-19 patients with severe neurological symptoms (neuro-COVID) as well as neuro-axonal damage in COVID-19 patients without severe neurological symptoms according to disease severity and after recovery, comparing the obtained findings with healthy donors (HD). Overall, COVID-19 patients (*n* = 55) showed higher plasma NfL levels compared to HD (*n* = 31) (*p* < 0.0001), especially those who developed ARDS (*n* = 28) (*p* = 0.0005). After recovery, plasma NfL levels were still higher in ARDS patients compared to HD (*p* = 0.0037). In neuro-COVID patients (*n* = 12), higher CSF and plasma NfL, and CSF MMP-2 levels in ARDS than non-ARDS group were observed (*p* = 0.0357, *p* = 0.0346 and *p* = 0.0303, respectively). SARS-CoV-2 RNA was detected in four CSF and two plasma samples. SARS-CoV-2 RNA detection was not associated to increased CSF NfL and MMP levels. During COVID-19, ARDS could be associated to CNS damage and alteration of BBB integrity in the absence of SARS-CoV-2 RNA detection in CSF or blood. CNS damage was still detectable after discharge in blood of COVID-19 patients who developed ARDS during hospitalization.

## 1. Introduction

The severe acute respiratory syndrome coronavirus 2 (SARS-CoV-2), which causes the coronavirus disease 2019 (COVID-19), has infected more than five hundred million people and has caused more than six million deaths globally, as of 5 July 2022 (available online: https://covid19.who.int, accessed on 5 July 2022). Whereas SARS-CoV-2 is known to cause mainly pulmonary diseases, including pneumonia and acute respiratory distress syndrome (ARDS), clinicians have observed many extrapulmonary manifestations of COVID-19 [1]. The emerging literature suggests that the hematologic, cardiovascular, renal, gastrointestinal, and hepatobiliary, endocrinologic, neurologic, ophthalmologic, and dermatologic systems can all be affected [2,3,4,5,6]. Several non-specific mild neurological symptoms can be observed in patients with COVID-19, including headache, anorexia, myalgia, fatigue, dizziness, anosmia, and ageusia [2,7,8], with variable incidence if we consider outpatients with milder presentations or inpatients with more severe disease forms [7,9,10]. In severe neurological cases of COVID-19, cerebrovascular accidents [11,12], confusion and impaired consciousness [13,14] can represent the clinical presentation of the disease. Acute inflammatory demyelinating polyneuropathy has also been reported in some patients [15,16]. In addition, meningoencephalitis [17], hemorrhagic posterior reversible encephalopathy syndrome [18], and acute necrotizing encephalopathy, involving the brainstem and basal ganglia, have been described in case reports [19,20]. Ocular manifestations, such as conjunctival congestion alone, conjunctivitis, and retinal changes, have also been reported in patients with COVID-19 [7,21,22,23].

SARS-CoV-2 RNA detection in the cerebrospinal fluid (CSF) represents the evidence of the ability of this virus to invade the central nervous system (CNS), although viral isolation in cellular culture represents the definitive sign of SARS-CoV-2 productive neuroinvasion [24]. In May 2020, Moriguchi and colleagues [19] were the first to report the presence of SARS-CoV-2 RNA in the CNS, using a real-time reverse transcription-polymerase chain reaction (RT-PCR) on a CSF sample of a patient with COVID-19 associated encephalopathy. Further cases have since been reported [25,26,27]. However, SARS-CoV-2 RNA was inconstantly detected in the CSF of COVID-19 patients with neurological manifestations [17,28,29].

In the management of neurological diseases, the identification and quantification of axonal damage can improve the diagnostic accuracy and prognostic assessment. Neurofilaments (Nf) are major components of the neuronal cytoskeleton, consisting predominantly of three subunits: Nf-light (NfL), Nf-medium, and Nf-heavy chains. Each subunit possesses a particular molecular mass, and their relative concentration is uneven. Upon neuro-axonal damage, NfL, the most abundant and soluble of the subunits [30], is released into the extracellular space and is detectable in the CSF and blood [31]. Its levels increase in CSF and blood proportionally to the degree of axonal damage in a variety of neurological disorders, including inflammatory, neurodegenerative, traumatic, cerebrovascular diseases, and in prion associated encephalopathy [32,33,34]. Evidence that both CSF and blood NfL may serve as diagnostic, prognostic, and monitoring biomarkers in neurological diseases are progressively accumulating, and NfL is one of the most promising biomarkers to be used in clinical and research settings in the near future [35].

Matrix metalloproteinases (MMPs) are zinc-dependent enzymes that degrade extracellular matrix (ECM) proteins, such as collagen, fibronectin, and laminin as well as basal membrane structures. MMPs are mediators of neuroinflammatory processes, by regulating blood–brain barrier (BBB) integrity, neutrophil infiltration in the CNS, and cytokine signaling [36]. The involvement of MMPs in the impairment of BBB integrity during viral infections of the CNS has been extensively reviewed elsewhere [37]. Clinical evidence from patients with viral meningitis have demonstrated elevated CSF levels of MMP-9 and tissue inhibitor of matrix metalloproteinases-1 (TIMP-1) compared to control patients [38]. Moreover, increased levels of chemokines, MMPs, and TIMP-1 have been reported in the CSF of patients with Varicella-zoster virus (VZV) infection and HIV infection [39,40], and in mice with lethal infection with neurotropic mouse hepatitis virus [41].

The primary aim of this study was to explore CSF and plasma NfL levels as well as CSF MMP-2 and MMP-9 levels in COVID-19 patients with severe neurological symptoms. The secondary aim was to evaluate in a cross-sectional and longitudinal approach plasma NfL levels in COVID-19 patients stratified according to disease severity during the acute phase of the disease and after recovery.

## 2. Materials and Methods

### 2.1. Patients and Clinical Samples Collection

Hospitalized COVID-19 patients were enrolled. COVID-19 related pneumonia was diagnosed by chest computed tomography (CT scan) associated with SARS-CoV-2 RNA detection on a nasopharyngeal swab, as previously described [6,42]. For all enrolled COVID-19 patients, whole blood samples were collected in heparin tubes during routine clinical testing on hospital admission (baseline) and after three months from hospital discharge (Tpost) (Figure 1A).

Patients were stratified according to the occurrence of severe neurological symptoms into two groups: with (neuro-COVID group) and without (COVID group) severe neurological symptoms. For the neuro-COVID group, CSF samples were collected in sterile tubes without anticoagulant whereas whole blood samples were collected in heparin and ethylenediaminetetraacetic acid (EDTA) tubes. Samples were drawn at the acute phase of the disease, during routine clinical testing (Figure 1A). Finally, healthy donors (HD) age and sex matched with COVID-19 patients, with negative nasopharyngeal swab for SARS-CoV-2 RNA detection and undetectable anti-SARS-CoV-2 Nucleoprotein (N) specific IgG on the sampling day, were enrolled (Figure 1A). Heparin and EDTA plasma samples were obtained after centrifugation and immediately stored at −80 °C until use. CSF samples were stored at -80 °C until use.

As reported in Figure 1B, COVID-19 patients were stratified according to disease severity into ARDS and non-ARDS groups. ARDS was defined according to the 2012 Berlin criteria [43]. Both neuro-COVID and COVID groups were further stratified according to disease severity (Figure 1B). Finally, neuro-COVID group was stratified based on the detection of SARS-CoV-2 RNA in CSF and plasma samples (Figure 1B), and the differences in CSF NfL levels as well as MMP-2 and MMP-9 activity were evaluated.

Finally, COVID-19 patients were further stratified into four groups according to the maximal oxygen supply/ventilation support required during the hospitalization: ambient air (AA), Venturi oxygen mask (VMK), noninvasive ventilation (NIV), and invasive mechanical ventilation through orotracheal intubation (IOT), and the differences in plasma NfL levels were evaluated.

### 2.2. Evaluation of CSF and Plasma NfL Levels in Collected Samples

As previously described [44], the evaluation of NfL levels in collected samples were assessed using the Simple PlexTM Ella Assay (ProteinSimple, San Jose, CA, USA) on EllaTM microfluidic system (Bio-Techne, Minneapolis, MN, USA) according to the manufacturers’ instructions. EllaTM was calibrated using the in-cartridge factory standard curve. The limit of detection of NfL was 1.09 pg/mL, calculated by adding three standard deviations to the mean background signal determined from multiple runs.

### 2.3. Evaluation of Gelatinase Activity by Zymography

For the neuro-COVID group, the evaluation of CSF MMP-2 and MMP-9 levels was performed by SDS-PAGE zymography as previously described [40,45]. Briefly, 50 μL of each CSF sample were precipitated for 1 h at −20 °C in cold acetone, then the samples were centrifuged, and the pellets solubilized in 10 μL of loading buffer containing SDS. Samples were then applied on 10% polyacrylamide gels (10 cm × 10 cm), which had been copolymerized with 0.1% (*w/v*) gelatin. Stacking gels contained 5.4% polyacrylamide. Electrophoresis was carried out at 4 °C for approximately 2 h at 100 V. After electrophoresis, the gels were washed for 2 × 30 min in 2.5% (*w/v*) Triton X-100 in 100 mM Tris-HCl, pH 7.4 (washing buffer) to remove SDS and reactivate the enzyme, then incubated for 24 h at room temperature in 100 mM Tris-HCl, pH 7.4 (developing buffer).

For the development of the enzyme activity, the substrate in the gels was stained with Coomassie brilliant blue R-250 and destained in methanol/acetic acid/H_2_O. Gelatinase activity was detected as a white band on a blue background and was quantified by computerized image analysis through two-dimensional scanning densitometry.

### 2.4. SARS-CoV-2 RNA Evaluation in CSF and Plasma Samples

For the neuro-COVID group, real-time RT-PCR and digital droplet PCR (ddPCR) were used for the detection and quantification of SARS-CoV-2 RNA in collected samples.

For real-time RT-PCR, total RNA was extracted from 1.5 mL of CSF and plasma using the Total Purification RNA kit (Norgen Biotek Corp., Thorold, ON, Canada) after virus concentration by centrifugation at higher speed, as previously described [46].

For SARS-CoV-2 RNA quantification by ddPCR, total RNA was extracted from 280 μL of samples using the QIAamp viral RNA mini kit (Qiagen, Hilden, Germany) according to manufacturer’s instruction and concentrated up to 10 μL by using Savant DNA SpeedVac (Thermo Fisher Scientific, Waltham, MA, USA). SARS-CoV-2 RNA was quantified by QX200TM Droplet DigitalTM PCR System (ddPCR, Biorad, Hercules, CA, USA) using an in-house assay, targeting the RdRP gene of SARS-CoV-2 as previously described [47,48]. The assay also targets the housekeeping gene RNAse P as internal control of amplification. The ddPCR assay provides an absolute quantification of viral RNA without a calibration curve and the results were expressed as copies/mL. Despite the standard curve, it is not necessary for proper DNA/RNA quantification; thus, the use of controls is very important, especially to better discriminate false positivity. As previously reported [47], our ddPCR assay for SARS-CoV-2 RNA quantification also shows a good linearity and reproducibility for the detection of a single RNA copy for reaction. At least 4 negative controls every 24 quantifications were used. The negative controls are treated as samples, starting from the extraction until the quantification, to exclude any potential contamination or procedure bias.

### 2.5. Statistical Analysis

All statistical analyses were performed using GraphPad Prism v.9 software and two-tailed *p* ≤ 0.05 was considered statistically significant. Values are represented as median and interquartile range (IQR). The nonparametric comparative Mann–Whitney test was used for comparing medians between COVID-19 patients and HD as well as between neuro-COVID and COVID groups and between ARDS and non-ARDS groups. The nonparametric comparative Wilcoxon test was used for longitudinal evaluation between biomarkers assessed at baseline and Tpost in the COVID group. Spearman rank correlation analysis was used to assess the relation between CSF and plasma NfL levels.

## 3. Results

### 3.1. Clinical and Demographical Feature of Study Population

Fifty-five hospitalized COVID-19 patients and 31 HD were enrolled (Table 1). According to chest CT scan, all COVID-19 patients had interstitial pneumonia and 52.7% had ARDS. Overall, 12.7% of COVID-19 patients died, and 21.8% of COVID-19 patients showed severe neurological symptoms (neuro-COVID group) (Table 1).

As reported in Table 2, in neuro-COVID group the most frequent neurological signs and symptoms included confusion and headache. Among less common symptoms, we observed a case of gaze deviation to the right, nystagmus, seizures, forced deviation of the head to the left, and bilateral vision impairment. Finally, a stroke, a meningoencephalitis and a bilateral optic neuritis were observed (Table 2). A CSF laboratory examination was reported in Table 3.

### 3.2. Evaluation of NfL in Study Population

Overall, all enrolled COVID-19 patients showed significantly higher plasma NfL levels compared to HD (27.1 [14.4–39.3] and 9.1 [5.7–12.4] pg/mL, *p* < 0.0001) (Figure 2A, Table 1). After stratifying COVID-19 patients into ARDS and non-ARDS groups, higher plasma NfL levels were observed in the ARDS compared to the non-ARDS group (33.8 [18.1–72.2] and 17.8 [10.2–27.6] pg/mL, respectively, *p* = 0.0005) (Figure 2B). Both ARDS and non-ARDS groups showed higher plasma NfL levels compared to HD (*p* < 0.0001 and *p* = 0.0212, respectively) (Figure 2B).

Interestingly, on hospital admission, we observed higher plasma NfL levels in patients that required oxygen support/ventilation during hospitalization (VMK, NIV and IOT) compared to HD (VMK: 26.7 [14.4–32.5], NIV: 29.4 [14.9–59.5], IOT: 39.0 [22.7–93.5]; *p* = 0.0258, *p* < 0.0001 and *p* < 0.0001, respectively) (Appendix A). Conversely, no differences were observed between the AA group and HD (14.8 [6.3–25.4]) (Appendix A). Finally, we observed that NIV and IOT groups showed significantly higher plasma NfL levels compared to the AA group (*p* = 0.0088 and *p* = 0.0023, respectively) (Appendix A).

COVID-19 patients were further stratified according to the presence comorbidities (Appendix A). On hospital admission, higher plasma NfL levels in patients with at least one comorbidity compared to those without was observed (29.2 [15.7–59.4] and 20.9 [11.2–29.9] pg/mL, respectively, *p* = 0.0436) (Appendix A). However, both groups (with and without comorbidities) showed higher plasma NfL levels compared to HD (*p* < 0.0001 and *p* = 0.0025, respectively) (Appendix A).

The longitudinal evaluation of plasma NfL levels performed in 38 COVID-19 patients showed a statistically significant decrease at the Tpost compared to baseline (13.8 [8.7–21.1] and 20.4 [11.6–30.2] pg/mL, respectively, *p* < 0.0001) (Figure 2C). At the Tpost, COVID-19 patients showed higher plasma NfL levels compared to HD, although the difference was not statistically significant (Figure 2C). After stratifying patients according to the occurrence of ARDS during hospitalization, at the Tpost, both ARDS and non-ARDS groups showed a reduction of plasma NfL levels compared to baseline (ARDS group: 28.1 [14.9–41.4] and 17.2 [12.6–23.3] pg/mL, respectively, *p* = 0.0095; non-ARDS group: 14.4 [8.9–26.7] and 9.7 [6.1–14.3] pg/mL, respectively, *p* = 0.0001) (Figure 2D and 2E, respectively). However, at the Tpost, plasma NfL levels were still significantly increased in ARDS group compared to HD (*p* = 0.0037) (Figure 2D), whereas no statistically significant differences were observed between non-ARDS group and HD (Figure 2E).

Stratifying COVID-19 patients according to the occurrence of severe neurological symptoms into neuro-COVID and COVID groups, plasma NfL levels were significantly increased in neuro-COVID compared to COVID group (71.7 [27.9–95.1] and 21.8 [13.9–34.0] pg/mL, respectively, *p* = 0.0034) (Figure 2F). Both neuro-COVID and COVID groups showed higher plasma NfL levels compared to HD (*p* < 0.0001 and *p* < 0.0001, respectively) (Figure 2F).

Stratifying neuro-COVID and COVID groups into ARDS and non-ARDS groups, we observed that both neuro-COVID ARDS and COVID ARDS groups showed higher plasma NfL levels than the corresponding non-ARDS groups (neuro-COVID group, ARDS vs. non-ARDS: 90.3 [71.7–99.9] and 27.2 [19.8–29.9] pg/mL, respectively, *p* = 0.0357; COVID group, ARDS vs. non-ARDS: 28.6 [15.3–42.6] and 14.8 [9.2–27.4] pg/mL, respectively, *p* = 0.0041) (Figure 2G). Interestingly, patients from the neuro-COVID ARDS group showed significantly increased plasma NfL levels compared to the COVID ARDS group (*p* = 0.0052) (Figure 2G).

Finally, the in neuro-COVID group, the evaluation of CSF NfL levels showed higher concentrations in the ARDS compared to the non-ARDS group (6480 [1512–11012] and 476 [305–2859] pg/mL, respectively, *p* = 0.0260) (Figure 3A). A positive correlation between NfL levels in CSF and plasma samples was observed (ρ = 0.8095 *p* = 0.0218).

### 3.3. CSF MMP Levels in Neuro-COVID Group

The zymographic analysis of CSF samples from neuro-COVID-19 patients evidenced on the gel two main lysis bands, present at different levels, with an apparent molecular mass of 72 and 92 kDa, corresponding to MMP-2 and MMP-9, respectively. The quantitative analysis of MMP levels in CSF samples by zymography showed significantly higher CSF MMP-2 levels in the ARDS compared to the non-ARDS group (90.5 [83.5–103.6] and 79.0 [61.4–85.7], respectively, *p* = 0.0303) (Figure 3B) and higher CSF MMP-9 levels, although not statistically significant (MMP-9: 20.5 [14.6–79.0] and 15.2 [10.6–17.8], respectively, *p* = 0.0823) (Figure 3C).

### 3.4. SARS-CoV-2 RNA Evaluation on CSF and Plasma Samples of Neuro-COVID Patients

For viral RNA detection, real-time RT-PCR and ddPCR were performed in collected CSF samples from neuro-COVID patients. Using RT-PCR, SARS-CoV-2 RNA was detected only in one CSF sample (Figure 4A and Table 3). Conversely, the evaluation of SARS-CoV-2 RNA with ddPCR showed viral RNA detection in the CSF of 4 out of 12 neuro-COVID patients and in plasma of 2 out of 8 neuro-COVID patients (Figure 4A and Table 3).

No statistically significant differences in CSF and plasma NfL levels, and CSF MMP-2 and MMP-9 levels, were observed after comparing neuro-COVID patients with and without SARS-CoV-2 RNA detection in CSF and plasma samples (Figure 4B–E).

## 4. Discussion

Evidence of the effects of SARS-CoV-2 on the CNS is evolving, with the virus being linked to neurological and psychiatric symptoms [17]. Several studies found COVID-19 to be associated with neurological manifestations in up to 36% of patients [49] and the most common reported manifestations were cerebrovascular events, followed by altered mental status [50]. Neurological manifestations can range from a mild headache or “brain fog” [51], to more serious complications such as Guillain-Barre syndrome [52], encephalitis [19], and arterial and venous strokes [53]. Several earliest reports of CNS involvement also described an unusually high rate of anosmia and dysgeusia [50]. The pathogenesis of these CNS effects of COVID-19 is still not known. In line with previous reports [54,55], in our study, we observed higher plasma NfL levels in COVID-19 patients on hospital admission compared to HD, especially in those who developed ARDS during hospitalization. A further stratification according to the maximal oxygen supply/ventilation support required during the hospitalization showed higher plasma NfL levels in patients who required NIV or IOT. Considering that NfL assessment was performed on hospital admission when patients were not yet subjected to ventilation, our data underline the potential role of plasma NfL levels as a prognostic marker of COVID-19 severity. These data are in line with Sutter et al. [55], showing that higher plasma NfL levels are associated with unfavorable short-term outcome in COVID-19 patients. Furthermore, in accordance with Aamodt et al. [56], and Masvekar et al. [57], the evaluation of plasma NfL levels on hospital admission might identify COVID-19 patients with either neurological comorbidities or increased risk of progression to severe COVID-19, thus requiring intensive cares, also focused in preventing further CNS injuries. The identification of a blood biomarker, such as plasma NfL, which is able to assess CNS impairment, could be useful to monitor the severity of the disease and optimize treatment strategies.

In our study, we found that COVID-19 patients with comorbidities showed higher plasma NfL levels on hospital admission compared to those without, although both groups showed higher plasma NfL levels compared to HD. It is not completely clear if comorbidities may impact plasma NfL levels. There is some evidence that plasma NfL levels could be influenced by body mass index (BMI) [58,59], renal function [60], and diabetes [61]. However, as previously showed by Koini et al. [62], the impact that comorbidities could have on plasma NfL levels is influenced by the age of the individual.

Increased plasma NfL levels were found in patients with severe neurological manifestations compared to patients without these symptoms, with the highest increase in neuro-COVID patients who developed ARDS compared to patients who did not. All these data argue in favor of a neuronal damage associated with COVID-19, especially in those patients with severe neurological symptoms and with a severe form of the disease. Furthermore, our data demonstrate increased plasma NfL levels in patients without neurological symptoms at the acute stage of COVID-19, suggesting the presence of subclinical CNS involvement in severe COVID-19 patients. Indeed, Nf is a structural protein that determines axonal caliber and conduction velocity in neurons [63]. One of the three Nf components, the NfL, has been proposed as a biomarker of axonal damage [64]. High CSF NfL levels have been found in patients with neurodegenerative conditions [65,66,67]. As a less invasive parameter, plasma NfL is supposed to be a surrogate marker, instead of CSF NfL in the evaluation of neural degeneration. Evidence that both CSF and blood NfL may serve as diagnostic, prognostic, and monitoring biomarkers in neurological diseases are progressively accumulating, and NfL is one of the most promising biomarkers to be used in clinical and research settings in the next future [68]. In our study, in line with previously report [35], a positive correlation between CSF and plasma NfL levels was observed, although performed in a small sample size. These data underline the potential role of plasma NfL evaluation to detect neuro-axonal injury and monitor COVID-19-associated neuronal damage.

The longitudinal evaluation of COVID-19 patients demonstrated that plasma NfL levels significantly decreased three months after discharge. Nevertheless, after recovery from the acute phase of the disease, plasma NfL levels were still altered in patients who developed ARDS during hospitalization, compared to HD. These data are in line with a recent report by Hampshire et al. [69], showing that severe COVID-19 illness is associated with objectively relevant measurable cognitive deficits that persist into the chronic phase. Thus, as already proven for multiple sclerosis, Alzheimer, and Parkinson disease [70,71], the persistence of high plasma NfL levels in patients with COVID-19 during the post-acute phase of the disease could represent a biomarker of neurocognitive impairment. In perspective, it will be useful to correlate plasma NfL levels with long-COVID clinical manifestations.

In our study, the high levels of CSF NfL, MMP-2, and MMP-9 observed in neuro-COVID with ARDS could be the expression of neuronal damage and BBB disruption possibly induced by the altered blood flow in the CNS and hypoxia. MMPs have been widely investigated for their role in the disruption of the BBB, particularly through the degradation of the components of the basal lamina, following stroke [72,73,74] and other cerebral pathologies such as traumatic brain injury [75]. However, a recent report has suggested that CSF MMP-9 levels were quantitatively linked to the amount of NfL release [76]. Indeed, MMPs are effector molecules of tissue damage that are released as a consequence of pro-inflammatory cytokine secretion [77,78]. MMPs can activate and regulate cytokine signaling in a positive feedback loop, enhancing the excessive inflammation [79,80,81]. This evidence underlines the role of MMPs not only in disrupting BBB integrity facilitating extravasation into the CNS, but also in promoting glial and neuronal cell death with the consequent increase in NfL levels. From a clinical point of view, the consequences of these mechanisms could be the development of neurological sequelae [82,83].

Our results are consistent with previous data showing that MMP-2 (more than MMP-9) plays a critical role in BBB disruption, glial cell activation, and white matter damages after chronic cerebral hypoperfusion in animal models and patients with cardiac arrest [84,85]. Indeed, in COVID-19 patients, CNS hypoxia can be a consequence of respiratory failure, thrombotic microangiopathy, and indirect effects of the vigorous inflammatory response with extensive cytokine activation. As suggested by Mohammadhosayni et al. [86], the high levels of inflammatory cytokines, such as TNF-α, might help to increase the production of MMPs, leading to BBB disruption in COVID-19 patients with neurological symptoms.

To date, is still unclear whether the amount of NfL crossing the BBB is dependent on the integrity of this barrier [87]. NfL is produced as a direct result of neuroaxonal injury but not as a direct result of BBB compromise per se. Impairment of BBB integrity can potentially facilitate NfL release out of the CNS into blood. However, as reported by other authors [88,89], in multiple sclerosis patients, where chronic inflammation leads to the disruption of the BBB, NfL does not systematically correlate with BBB integrity. Mechanistic studies are needed to establish causative precedence but is plausible that the loss of integrity of the BBB increases permeability of pre-existing NfL from CNS into blood.

Finally, in this manuscript, we have also addressed the issue of SARS-CoV-2 RNA detection in CSF and plasma samples of patients with severe neurological symptoms. These results were obtained using ddPCR, a highly sensitive method for nucleic acids quantification, compared to real-time RT-PCR. In the clinical practice, the gold standard for the detection and quantification of SARS-CoV-2 RNA is the real-time RT-PCR. Recently, attention has been focused on the use of ddPCR system for the detection and quantification of SARS-CoV-2 RNA. This assay provides a reliable absolute quantification of viral RNA and is endowed with a higher sensitivity compared to real-time RT-PCR, specifically for quantifying low viral loads [90,91]. Indeed, in our study, only one out of twelve CSF sample showed SARS-CoV-2 RNA detection using real-time RT-PCR, whereas four out of twelve CSF samples showed SARS-CoV-2 RNA positivity using ddPCR. Moreover, the RT-PCR positive CSF sample showed the highest viral load when assessed with the ddPCR. Of note, for the four patients with SARS-CoV-2 RNA ddPCR detection in the CSF, the corresponding plasma samples tested negative (using the same molecular method), therefore, a viral carry-over from the blood to the CNS seems unlikely. From the comparison of the two methods, as expected, we observed a better performance of ddPCR respect to real-time RT-PCR, suggesting that ddPCR can represent an added value in reducing false negative results and in detecting very low concentrations of viral RNA. These characteristics can represent a useful tool for better identifying viral dissemination in extra-pulmonary regions and in turn can unravel its significance in terms of virus transmissibility and extra-pulmonary clinical manifestations. Even if the in vitro replication of SARS-CoV-2 in neural cells has been widely reported [92,93,94], a limit of ddPCR (and also of RT-PCR) is that the detection of SARS-CoV-2 RNA is not equal to identification of effectively infectious viral particles, so these methods are not informative about active viral replication but only on the presence of viral RNA.

In our cohort, no statistically significant differences were observed in CSF NfL levels as well as in CSF MMP-2 and MMP-9 levels after stratifying patients with severe neurological symptoms according to SARS-CoV-2 RNA detection by ddPCR on either CSF or plasma samples. This aspect, together with the increased NfL levels observed in ARDS patients, suggests that CNS damage is prevalently associated with COVID-19 severity rather than SARS-CoV-2 RNA detection. This is in line with the evidence obtained by other authors, showing that neurological signs and symptoms could be the effect of hyperinflammation and hypoxia [91]. Nevertheless, our data evidenced the ability of SARS-CoV-2 to invade the CNS. SARS-CoV-2 infection of human host cells is mediated mainly by the cellular receptor ACE-2, which is expressed at very low levels in the CNS under normal conditions [95]. To date, several reports on potential neuroinvasion by SARS-CoV-2 appeared in the literature although most of them were conducted on animal models [94,96,97]. We should also consider that the presence of viral RNA does not directly correspond to the presence of viable viral particles, which should be assessed with viral isolation in cultured cells.

In summary, our data suggest CNS damage in COVID-19 patients during the acute phase of the disease, which is mainly related to COVID-19 severity rather than SARS-CoV-2 neuro-invasion. COVID-19 patients who developed ARDS during the acute phase of the disease tended to maintain higher levels of NfL compared to HD, up to three months after hospital discharge. Measurement of NfL levels in plasma samples represents a convenient and easy to perform method to assess neuronal damage in the context of COVID-19. As NfL is specific for neuronal damage, the increased plasma NfL levels in patients without neurological symptoms suggest the presence of subclinical CNS involvement in COVID-19 patients, especially in those with the most severe forms of the disease. The implications of this subclinical CNS involvement need to be further elucidated and correlated with long-COVID manifestations.

## 5. Conclusions

A growing number of COVID-19 patients showed neurologic symptoms during the acute stage of the disease as well as several neurological sequelae following COVID-19 recovery. To our knowledge, this is the first study providing a CNS damage and alteration of BBB integrity in COVID-19 patients with severe neurological symptoms during the acute phase of the disease, which is mainly related to COVID-19 severity rather than SARS-CoV-2 neuro-invasion. COVID-19 patients who developed ARDS during the acute phase of the disease tended to maintain high levels of NfL, up to three months after hospital discharge. Measurement of NfL levels in plasma samples represents a convenient and easy to perform method to assess neuronal damage in the context of COVID-19.

## Figures and Tables

**Figure 1 cells-11-02480-f001:**
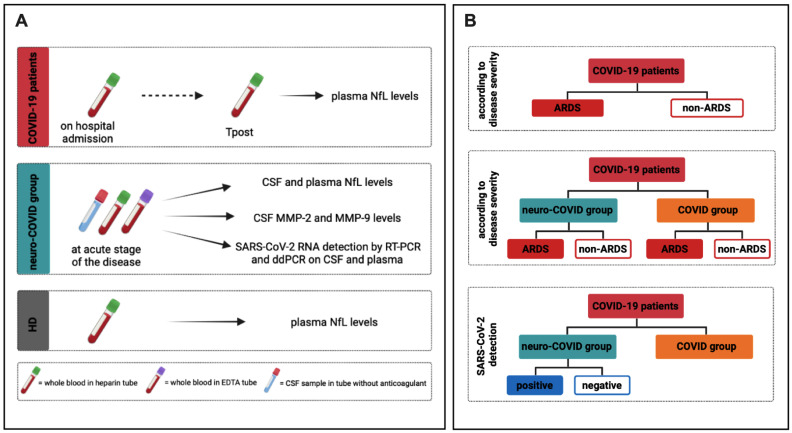
Study design. (**A**) For COVID-19 patients, whole blood samples were collected in heparin tubes during routine clinical testing at two timepoints: on hospital admission (baseline) and after three months from discharge (Tpost). For the neuro-COVID group, CSF samples and heparin and EDTA whole blood samples were collected. RT-PCR and ddPCR were used for the detection and quantification of SARS-CoV-2 RNA in collected CSF and whole blood samples. For healthy donors (HD) heparin whole blood samples were collected. (**B**) According to clinical outcome, COVID-19 patients were stratified into ARDS and non-ARDS groups and the differences in plasma NfL levels were evaluated. According to disease severity, both neuro-COVID group and COVID groups were stratified into ARDS and non-ARDS groups and the differences in CSF NfL levels, plasma NfL levels, and CSF MMP-2 and MMP-9 levels were assessed. According to the detection of SARS-CoV-2 RNA in CSF and plasma samples, neuro-COVID group was stratified into positive and negative groups and the differences in CSF NfL levels as well as MMP-2 and MMP-9 levels were evaluated. Neuro-COVID group: COVID-19 patients with severe neurological symptoms; COVID group: COVID-19 patients without severe neurological symptoms; NfL: neurofilament light chain; MMP-2: matrix metalloprotease-2; MMP-9: matrix metalloprotease-9; SARS-CoV-2: severe acute syndrome coronavirus 2; RT-PCR: reverse transcription-polymerase chain reaction; ddPCR: droplet digital polymerase chain reaction; CSF: cerebrospinal fluid; HD: healthy donors; EDTA: ethylenediaminetetraacetic acid; ARDS: acute respiratory distress syndrome.

**Figure 2 cells-11-02480-f002:**
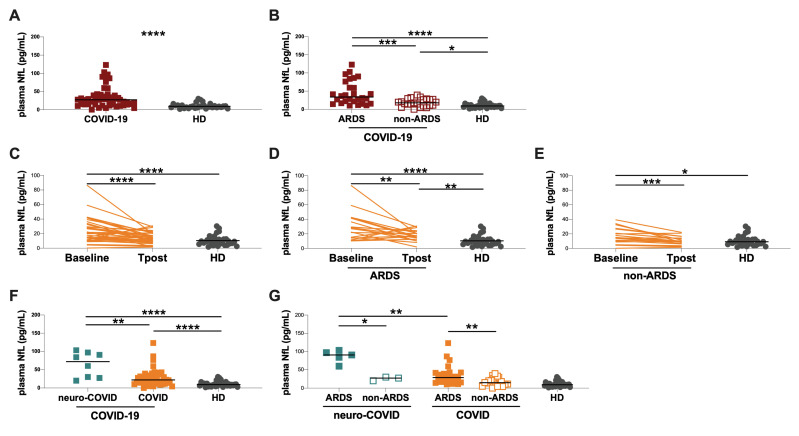
Evaluation of plasma NfL levels in COVID-19 patients and healthy donors. (**A**) Evaluation of plasma NfL levels in COVID-19 patients and HD. (**B**) Evaluation of plasma NfL levels in patients with ARDS (ARDS group), patients without ARDS (non-ARDS group) and HD. (**C**) Longitudinal evaluation of plasma NfL levels in COVID-19 patients. (**D**) Longitudinal evaluation of plasma NfL levels in COVID-19 patients who developed ARDS during hospitalization. (**E**) Longitudinal evaluation of plasma NfL levels in COVID-19 patients who did not developed ARDS during hospitalization. (**F**) Evaluation of plasma NfL levels in COVID-19 patients with severe neurological symptoms (neuro-COVID group), COVID-19 patients without severe neurological symptoms (COVID group) and HD. (**G**) Evaluation of plasma NfL levels in neuro-COVID and COVID groups stratified according to ARDS. Horizontal lines represent medians. COVID-19: coronavirus disease 2019; neuro-COVID group: COVID-19 patients with severe neurological symptoms; COVID group: COVID-19 patients without severe neurological symptoms; NfL: neurofilament light chain; HD: healthy donors; ARDS: acute respiratory distress syndrome; CSF: cerebrospinal fluid. ****: *p* < 0.0001; ***: 0.0001 < *p* < 0.001; **: 0.001 < *p* < 0.01; *: 0.01 < *p* < 0.05.

**Figure 3 cells-11-02480-f003:**
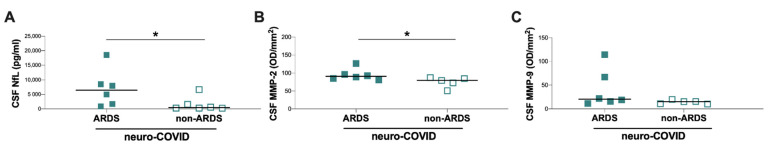
Evaluation of CSF NfL, MMP-2, and MMP-9 levels in neuro-COVID group. (**A**) Evaluation of CSF NfL levels in COVID-19 patients with severe neurological symptoms (neuro-COVID group) stratified according to ARDS development. (**B**) Evaluation of CSF MMP-2 levels in COVID-19 patients with neurological symptoms (neuro-COVID group) stratified according to ARDS development. (**C**) Evaluation of CSF MMP-9 levels in COVID-19 patients with neurological symptoms (neuro-COVID group) stratified according to ARDS development. Horizontal lines represent medians. Neuro-COVID group: COVID-19 patients with severe neurological symptoms; NfL: neurofilament light chain; HD: healthy donors; ARDS: acute respiratory distress syndrome; CSF: cerebrospinal fluid; MMP-2: matrix metalloprotease-2; MMP-9: matrix metalloprotease-9; OD: optical density. *: 0.01 < *p* < 0.05.

**Figure 4 cells-11-02480-f004:**
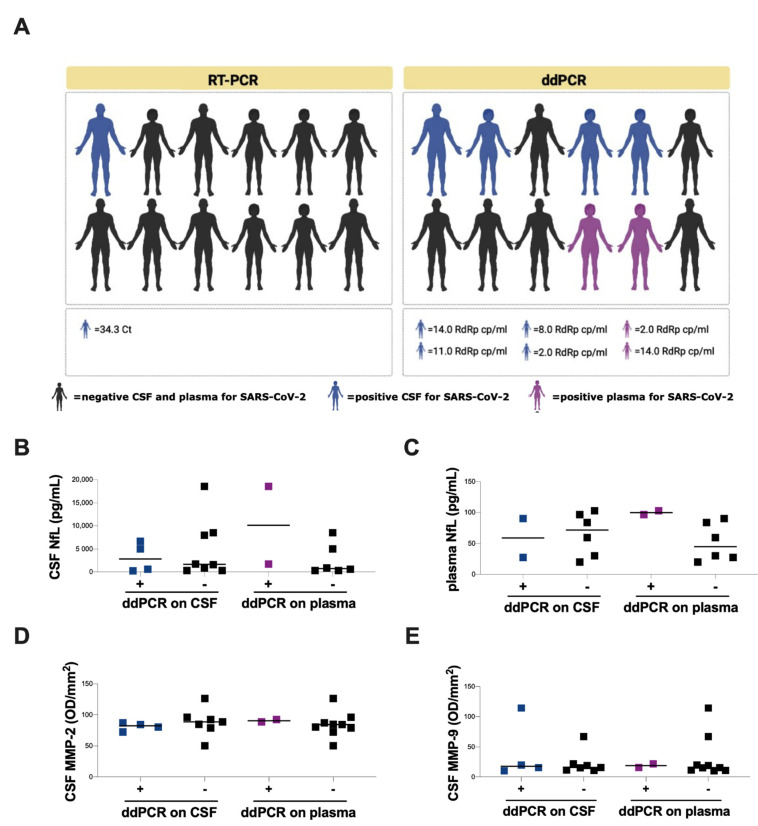
Detection of SARS-CoV-2 RNA on CSF and plasma samples and evaluation of NfL and MMPS levels in neuro-COVID group. (**A**) Among COVID-19 patients with neurological symptoms (neuro-COVID group), the detection of SARS-CoV-2 RNA on CSF and plasma samples was performed using RT-PCR and ddPCR. (**B**) Evaluation of CSF NfL levels in COVID-19 patients with severe neurological symptoms (neuro-COVID group) stratified according to ddPCR positivity on CSF and plasma. (**C**) Evaluation of plasma NfL levels in COVID-19 patients with severe neurological symptoms (neuro-COVID group) stratified according to ddPCR positivity on CSF and plasma. (**D**) Evaluation of CSF MMP-2 levels in COVID-19 patients with severe neurological symptoms (neuro-COVID group) stratified according to ddPCR positivity on CSF and plasma. (**E**) Evaluation of CSF MMP-9 levels in COVID-19 patients with severe neurological symptoms (neuro-COVID group) stratified according to ddPCR positivity on CSF and plasma. Horizontal lines represent medians. SARS-CoV-2: severe acute respiratory syndrome coronavirus 2; RT-PCR: reverse transcription-polymerase chain reaction; ddPCR: droplet digital polymerase chain reaction; CSF: cerebrospinal fluid; NfL: neurofilament light chain; MMP-2: matrix metalloprotease-9; MMP-9: matrix metalloprotease-9.

**Table 1 cells-11-02480-t001:** Demographic and clinical features in COVID-19 patients.

	COVID-19 Patients (*n* = 55)	HD (*n* = 31)	*p* Value *
Male/Female	32/23	15/16	ns
Age, years	63 (55–73)	64 (55–70)	ns
ARDS/non-ARDS	26/29	-	-
Deaths/Alive	7/48	-	-
**Comorbidities**			
Any	32	-	-
Hypertension	19	-	-
Cardiovascular	4	-	-
Diabetes	6	-	-
Pulmonary	4	-	-
Cancer	6	-	-
Renal	1	-	-
**Symptoms**			
Fever	44	-	-
Cough	26	-	-
Shortness of breath	19	-	-
Myalgia or arthralgia	14	-	-
Neurological symptoms	12		
Diarrhea	7	-	-
Anosmia and ageusia	4	-	-
Sputum production	1	-	-
**Laboratory finding**			
WBC (×10^9^/L)	4.9 (4.2–5.8)	-	-
Neutrophils (×10^9^/L)	3.5 (2.3–4.1)	-	-
Lymphocytes (×10^9^/L)	1.1 (0.7–1.5)	-	-
NLR	2.9 (1.6–5.2)	-	-
CRP (mg/dL)	3.4 (1.3–11.7)	-	-
D-dimer (μg/mL)	823 (443–1702)	-	-
Ferritin (ng/mL)	493 (264–1445)	-	-
LDH (U/L)	260 (201–354)	-	-
P/F ratio	343 (293–407)	-	-

HD: healthy donors, *n*: number, ARDS: acute respiratory distress syndrome, WBC: white blood cells, NLR: neutrophil/lymphocyte ratio, CRP: C-reactive protein, LDH: lactate dehydrogenase, P/F: arterial oxygen partial pressure/fraction of inspired oxygen. Data are shown as median (interquartile range). * The 2-tailed X2 test or Fisher’s exact test and the nonparametric comparative Mann–Whitney test were used for comparing proportions and medians, respectively, between COVID-19 patients and HD.

**Table 2 cells-11-02480-t002:** Demographic and clinical features of neuro-COVID group.

Patient	Gender	Age	Comorbidities	Neurologic Signs and Symptoms	Real-Time RT-PCR in Nasopharyngeal Swab (Ct Values)	Neurological Outcomes	Outcome
**1**	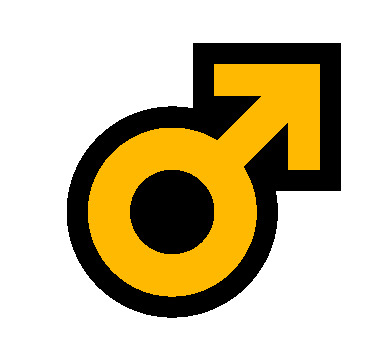	67	-	confusion	positive (20.4)		discharged
**2**	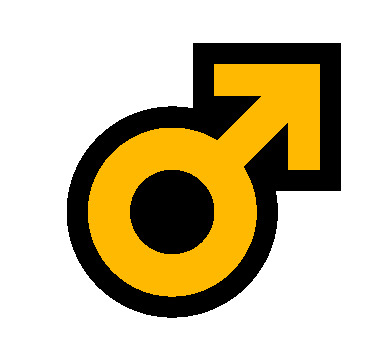	83	diabetes mellitus (type II)	confusion, syncope	positive (30.9)		discharged
**3**	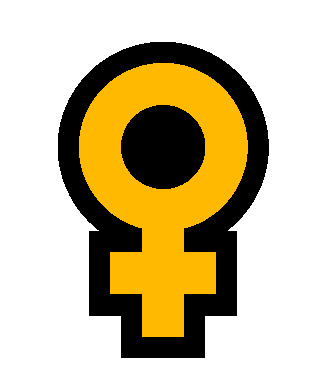	70	arterial hypertension and chronic lymphoid leukemia	headache, confusion	positive (29.5)		discharged
**4**	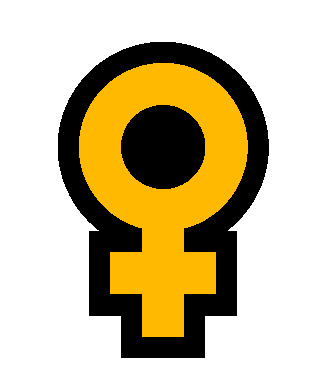	61	dyslipidemia	nystagmus, seizure, forced deviation of the head to the left	positive (n.a.)		discharged
**5**	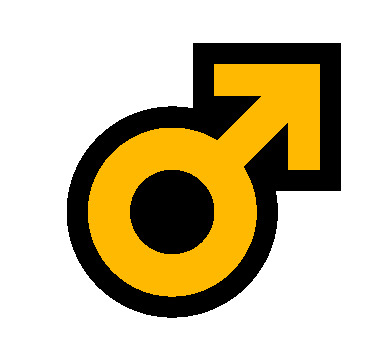	86	diabetes mellitus (type II), dyslipidemia, arterial hypertension	weakness, headache, gaze deviation to the right	positive (n.a.)	stroke	discharged
**6**	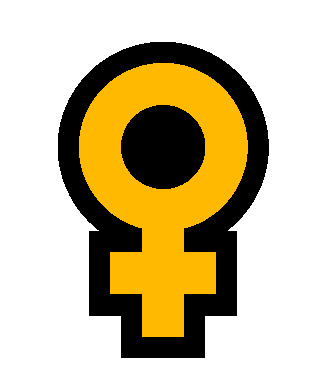	58	bronchial asthma	headache, confusion	positive (24.7)		discharged
**7**	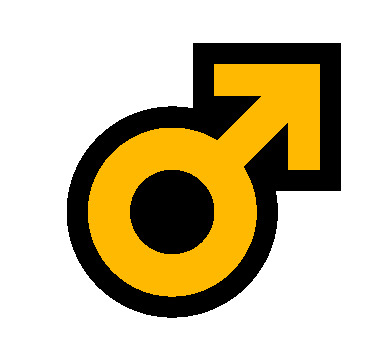	36	acute myeloid leukemia	headache, confusion	positive (10.4)	meningoencephalitis	death
**8**	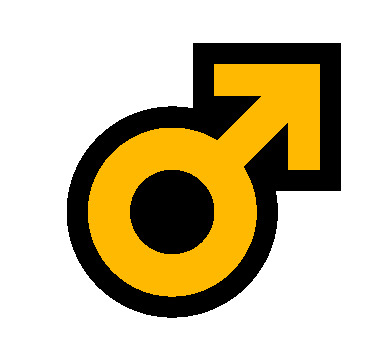	69	-	lower limb paresthesia	positive (14.0)		discharged
**9**	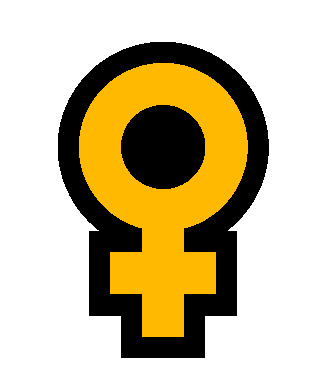	67	bronchial asthma, arterial hypertension, right nephrectomy, anemia	headache, confusion	positive (15.7)		discharged
**10**	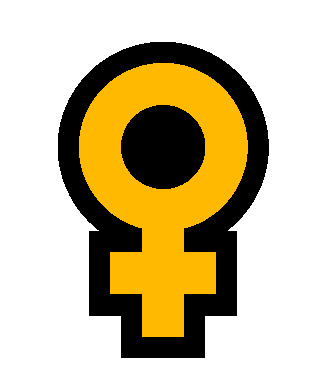	62	diabetes mellitus (type II), dyslipidemia, arterial hypertension	impaired bilateral vision and frontal headache	positive (32.3)	bilateral optic neuritis	discharged
**11**	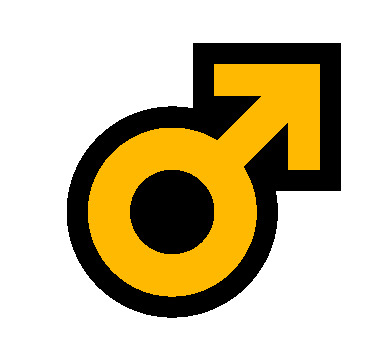	78	arterial hypertension, diabetes mellitus (type II)	headache, confusion	positive (28.8)		death
**12**	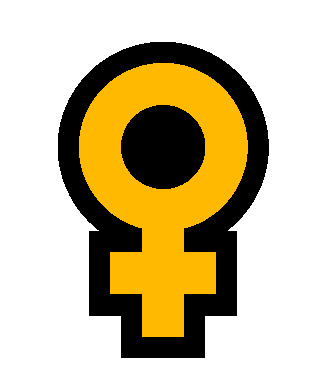	60	solid tumor	headache, confusion	positive (24.1)		discharged

RT-PCR: reverse transcription-polymerase chain; Ct: cycle threshold; n.a.: not available.

**Table 3 cells-11-02480-t003:** CSF examination of neuro-COVID group.

Patient	Appearance	Protein mg/dL [15–45]	Glucose mg/dL [50–80]	Albumin mg/dL [0–35]	Lactic Acid mmol/L [1.1–2.4]	Cell Count/μL [<10]	QAlb [0–9]	Real-Time RT-PCR in CSF (Ct Values)	Real-Time RT-PCR in Plasma	ddPCR in CSF (RdRp cp/mL)	ddPCR in Plasma (RdRp cp/mL)
**1**	clear	28	80	8.2	1.1	2	3.3	negative	negative	negative	negative
**2**	clear	34	104	9.1	2.1	1	4.3	negative	negative	negative	negative
**3**	clear	17.2	76	9.3	1.4	1	2.6	negative	negative	positive (11.0)	negative
**4**	clear	18.2	89	8.9	1.6	1	3.1	negative	negative	negative	negative
**5**	clear	28.9	98	8.3	3.2	2	2.5	negative	negative	negative	negative
**6**	clear	37.0	64	9.1	3.4	5	4.0	negative	negative	negative	negative
**7**	clear	19.6	61	10.5	3.3	16	3.6	negative	negative	positive (8.1)	negative
**8**	clear	39.6	56	20.9	3.3	3	6.1	negative	negative	negative	positive (2.0)
**9**	clear	55.8	101	35.9	3.4	4	9.4	negative	negative	positive (2.0)	negative
**10**	clear	29.4	151	17.9	3.1	5	5.0	negative	negative	negative	negative
**11**	clear	25.0	71	10.0	2.3	1	4.3	negative	negative	negative	positive (14.0)
**12**	clear	47.0	86	29.9	1.8	5	7.4	positive (34.3)	negative	positive (14.0)	negative

QAlb: CSF/serum albumin quotient; RT-PCR: reverse transcription-polymerase chain reaction; Ct: cycle threshold, ddPCR: droplet digital polymerase chain reaction; RdRp: RNA-dependent RNA polymerase; cp: copy.

## Data Availability

All data generated or analyzed during this study are included in this published article. The datasets used and/or analyzed during the current study are available from the corresponding author on reasonable request.

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
