# Peer review of "Neuro-Axonal Damage and Alteration of Blood–Brain Barrier Integrity in COVID-19 Patients"

_cells, 2022, doi:10.3390/cells11162480_

Round 1

Reviewer 1 Report

The manuscript of Zingaropoli et al. with the title „Neuro-axonal damage and alteration of blood-brain barrier integrity in COVID-19 patients“ investigates the level on NfL in serum of COVID-19 patients and healthy controls. They describe the upregulation of NfL in serum of COVID-19 patients. This is even more significant when they stratify for patients with ARDS. This is also the case in patients with neurologic symptoms. Moreover, the group is also assessing the levels of NfL, MMP-2 and MMP-9 in CSF of COVID-19 patients. In the latter part of the study, no control patients were included.

I have several concerns:

While I find this study interesting, I am wondering whether the increased levels of NfL might be due to the fact that the patients with COVID-19 included in the study might just have more co-morbidities such as hypertension, which might increase the NfL levels. Thus, please include a novel table to display the single patients, their individual co-morbidities, and the respective NfL load in serum and CSF. Discuss the possibility that the co-morbidities specifically in the COVID-19 group may have influenced the results.

The same concern applies for the ARDS-groups. An increase in NfL might be simply the consequence of lack of oxygen or ventilation. The latter is prone to change a couple of serum parameters. Please also include in the aforementioned new table, if patients were on ventilation or not. Please discuss the possibility in the discussion that ventilation might have influenced the study outcome. Cite relevant papers.

Interestingly, neither NfL nor MMP amounts seem to be dependent on viral abundance, which was only determined in the “neuro” COVID group. Here the authors used conventional RT-PCR to determine viral loads in serum and CSF. Only one patient showed a positive deltaCT of 34.3, which is very low and below the threshold that we use in our laboratory. The authors also applied an in-house method for detection of viral nuclei acids ddPCR. As with any super sensitive method, I see the risk of false positive results, here. The measured values seem very low. Thus, please include ddPCRs of serum of the here included control patients to define a threshold for false positive values.

Author Response

The manuscript of Zingaropoli et al. with the title “Neuro-axonal damage and alteration of blood-brain barrier integrity in COVID-19 patients” investigates the level on NfL in serum of COVID-19 patients and healthy controls. They describe the upregulation of NfL in serum of COVID-19 patients. This is even more significant when they stratify for patients with ARDS. This is also the case in patients with neurologic symptoms. Moreover, the group is also assessing the levels of NfL, MMP-2 and MMP-9 in CSF of COVID-19 patients. In the latter part of the study, no control patients were included.

I have several concerns:

While I find this study interesting, I am wondering whether the increased levels of NfL might be due to the fact that the patients with COVID-19 included in the study might just have more co-morbidities such as hypertension, which might increase the NfL levels. Thus, please include a novel table to display the single patients, their individual co-morbidities, and the respective NfL load in serum and CSF. Discuss the possibility that the co-morbidities specifically in the COVID-19 group may have influenced the results.

As suggested by the Referee #1, we included a new table (Supplementary Table 1), showing plasma and CSF NfL levels on hospital admission, the comorbidities, as well as the maximal ventilation needed during hospitalization for all the enrolled patients. As reported on page 8, line 240, stratifying COVID-19 patients according to the presence of at least one comorbidity, on hospital admission, higher plasma NfL levels were observed in patients with comorbidities compared to patients without (Supplementary Figure 1B). However, both groups (with and without comorbidities) showed higher plasma NfL levels compared to HD. As reported in the Discussion section (page 12, line 358), it is not completely clear if and how comorbidities may impact plasma NfL levels. There is some evidence that plasma NfL levels may be influenced by body mass index (BMI) (Manouchehrinia et al. 2020; Nilsson et al. 2019), renal function (Akamine et al. 2020) and diabetes (Thota et al. 2022). However, as previously showed by Koini et al. (Koini et al. 2021), the impact that comorbidities could have on plasma NfL levels is influenced by the age of the individual.

The same concern applies for the ARDS-groups. An increase in NfL might be simply the consequence of lack of oxygen or ventilation. The latter is prone to change a couple of serum parameters. Please also include in the aforementioned new table, if patients were on ventilation or not. Please discuss the possibility in the discussion that ventilation might have influenced the study outcome. Cite relevant papers.

As suggested by the Referee #1, patients were further stratified into four groups according to the maximal oxygen supply/ventilation support required during the hospitalization: ambient air (AA), Venturi oxygen mask (VMK), noninvasive ventilation (NIV) and invasive mechanical ventilation through orotracheal intubation (IOT). As reported in the Results section (page 8, line 231), on hospital admission we observed higher plasma NfL levels in patients requiring oxygen support and ventilation (VMK, NIV and IOT) during hospitalization compared to patients that did not need oxygen support nor ventilation and HD (Supplementary Figure 1A). Moreover, no differences were observed between AA group and HD (Supplementary Figure 1A). We observed that VMK, NIV and IOT groups showed higher plasma NfL level compared to AA group (Supplementary Figure 1A). Considering that NfL assessment was performed on hospital admission when ventilation was not yet started, our data underline the potential role of plasma NfL levels as a prognostic marker of COVID-19 severity. Indeed, as evidenced in the Discussion section (page 12, line 344), our results are in line with Sutter et al. (Sutter et al. 2021), showing that higher plasma NfL levels are associated with unfavorable short-term outcome in COVID-19 patients. Furthermore, in accordance with Aamodt et al. (Aamodt et al. 2021), and Masvekar et al. (Masvekar et al. 2022), the evaluation of plasma NfL levels on hospital admission might identify COVID-19 patients with either neurological comorbidities or with an increased risk of progression to severe COVID-19, thus requiring intensive cares, also focused in preventing further CNS injuries. The identification of a blood biomarker, such as plasma NfL, able to assess CNS impairment could be useful to monitor the severity of the disease and optimize treatment strategies.

Interestingly, neither NfL nor MMP amounts seem to be dependent on viral abundance, which was only determined in the “neuro” COVID group. Here the authors used conventional RT-PCR to determine viral loads in serum and CSF. Only one patient showed a positive deltaCT of 34.3, which is very low and below the threshold that we use in our laboratory. The authors also applied an in-house method for detection of viral nuclei acids ddPCR. As with any super sensitive method, I see the risk of false positive results, here. The measured values seem very low. Thus, please include ddPCRs of serum of the here included control patients to define a threshold for false positive values.

As correctly mentioned by the Referee #1, the droplet ddPCR is a highly sensitive method for nucleic acids quantification. As reported in Material and Methods section (page 5, line 182), this method allows an absolute quantification without a calibration curve. Despite the standard curve is not necessary for the proper DNA/RNA quantification, the use of controls is very important, especially to better discriminate false positivity. As reported by Alteri C. et al. (Alteri et al. 2020), the ddPCR assay for SARS-CoV-2 RNA quantification shows a good linearity and reproducibility also for the detection of a single RNA copy for reaction. In this light, in our experience, we use to quantify at least 4 negative controls every 24 quantifications or at least for each single run if the quantifications are lower. The negative controls are treated as samples, starting from the extraction until the quantification, to exclude any potential contamination or procedure bias. By re-analyzing all negative controls of each SARS-CoV-2 RNA ddPCR run performed, we found that for a total of 6 runs, we quantified 26 negative controls (approximatively 4 negative controls per run) and in only one reaction we observed an event of weak increased fluorescence in comparison to all other negative droplets, even if under the positivity cut-off. Moreover, in our experience, by analyzing within another study population, 40 plasma samples from 40 hospitalized SARS-CoV-2 infected patients without neurological implications, we found detectable SARS-CoV-2 RNA in plasma in 12 out of 40, with a median (IQR) of 14 (11-21) cp/ml, suggesting how a low detection of RNAemia occurs in a small fraction of infected patients.

Reviewer 2 Report

The authors present a timely study on neurological impact from SARS-CoV-2 infection.  I believe this information will be of interest and informative to a broad readership, but a few questions or concerns should be addressed first:

1)     Very minor edit, but the authors use the term “next future” a few times, such as line 83.  Do they mean near future?

2)     ddPCR vs RTPCR detection discrepancy of SARS-Cov-2 is not really explained or discussed much, please expand on this discrepancy

3)     was imaging (i.e MRI) or any other assessement done to evaluate BBB integrity or was MMP the only marker used?

4)     It’s a bit unclear, is there a direct link between NfL and MMP or did the authors really just have 2 different objectives with this report?

5)     It appears the HD values are not on the plots, is there a reason why?  It would be helpful to have those on the plots for comparison

6)     The plot in 3B isn’t really informative as presented and can be a bit confusing

7)     Are the authors saying that presence of NfL in CSF and plasma is from loss of BBB integrity?

Author Response

The authors present a timely study on neurological impact from SARS-CoV-2 infection.  I believe this information will be of interest and informative to a broad readership, but a few questions or concerns should be addressed first:

  • Very minor edit, but the authors use the term “next future” a few times, such as line 83. Do they mean near future?

As suggested by the Referee #2, we modified “next future” with “near future” on page 2, line 83.

  • ddPCR vs RTPCR detection discrepancy of SARS-Cov-2 is not really explained or discussed much, please expand on this discrepancy

As suggested by Referee #2, we expanded the real-time RT-PCR and ddPCR discrepancy in the Discussion section on page 13, line 442. From the comparison of the two methods, as expected, we observed a better performance of ddPCR respect to real-time RT-PCR, suggesting that ddPCR can represent an added value in reducing false negative results and in detecting very low concentrations of viral RNA. These characteristics can represent a useful tool for better identify viral dissemination in extra-pulmonary regions and in turn can unraveling its significance in terms of virus transmissibility and extra-pulmonary clinical manifestations. Even if the in vitro replication of SARS-CoV-2 in neural cells has been widely reported (Ramani et al. 2020; Pellegrini et al. 2020; Song et al. 2021) a limit of ddPCR (and also of RT-PCR) is that the detection of SARS-CoV-2 RNA is not equal to identification of effectively infectious viral particles, so these methods are not informative about active viral replication but only on the presence of viral RNA.

  • was imaging (i.e MRI) or any other assessement done to evaluate BBB integrity or was MMP the only marker used?

Unfortunately, in our cohort nor MRI neither other CNS imaging techniques were performed, mainly because of COVID-19 restrictions in performing not strictly necessary investigations, especially during the first and second “waves” of 2020, when most of the patients were enrolled in the present study. Therefore, in our study, the assessment of MMP activity in the CSF was the only way used to evaluate possible BBB alterations in COVID-19 patients with neurological symptoms. In this regard, as reported on page 13, line 400, it is important to underline that MMPs have been widely investigated for their role in the disruption of the BBB, particularly through the degradation of extracellular matrix (ECM) components and tight junction proteins (Rempe, Hartz, e Bauer 2016), following stroke (Romanic et al. 1998; Rosenberg 2002; Gidday et al. 2005) and other cerebral pathologies such as traumatic brain injury (Planas, Solé, e Justicia 2001).

  • It’s a bit unclear, is there a direct link between NfL and MMP or did the authors really just have 2 different objectives with this report?

A possible link between MMPs and NfL was already hypothesized. Indeed, in a recent paper (Le et al. 2020), CSF MMP-9 levels have been quantitatively linked to the amount of NfL release. We now have added a sentence in the Discussion section (page 13, line 405) explaining that MMPs are effector molecules of tissue damage which are released as a consequence of pro-inflammatory cytokine secretion (Vecil et al. 2000; Reinhard, Razak, e Ethell 2015). MMPs can activate and regulate cytokine signaling in a positive feedback loop, enhancing the excessive inflammation (Sellner e Leib 2006; Muri et al. 2019; Schönbeck, Mach, e Libby 1998). This evidence underlines the role of MMPs not only in disrupting BBB integrity facilitating extravasation into the CNS, but also in promoting glial and neuronal cell death with the consequent increase in NfL levels. On a clinical point of view, the consequences of these mechanisms could be the development of neurological sequelae (Anthony et al. 1998; Leppert et al. 2000).

  • It appears the HD values are not on the plots, is there a reason why? It would be helpful to have those on the plots for comparison

As suggested by the Referee #2, HD values were added in Figure 2G. Unfortunately, for Figure 3 and 4, HD values were not available considering that lumbar puncture is an invasive procedure and was performed in patients with specific clinical indications. The commonest indications to performed lumbar puncture are represented by the sudden onset of neurological symptoms, with the evidence of inflammatory and infectious etiologies (Doherty e Forbes 2014). Therefore, lumbar puncture was not performed in HD.

  • The plot in 3B isn’t really informative as presented and can be a bit confusing

As suggested by the Referee #1, we removed the plot 3B.

  • Are the authors saying that presence of NfL in CSF and plasma is from loss of BBB integrity?

To date, is still unclear whether the presence of NfL in CSF and plasma is dependent on the integrity of BBB barrier (Barro, Chitnis, e Weiner 2020). As reported on page 13, line 421, NfL is produced as a direct result of neuroaxonal injury but not as a direct result of BBB compromise per se. Impairment of BBB integrity can potentially facilitate NfL release out of the CNS into blood. However, as reported by other authors (Kalm et al. 2017; Uher et al. 2021), in multiple sclerosis patients where chronic inflammation leads to a disruption of the BBB, NfL does not systematically correlate with BBB integrity. Mechanistic studies are needed to establish causative precedence but is plausible that the loss of integrity of the BBB increases permeability of pre-existing NfL from CNS into blood.

Round 2

Reviewer 1 Report

Thank you so much for the revision of your manuscript. All my concerns have been sufficiently addressed.